# Coagulation Abnormalities in Renal Pathology of Chronic Kidney Disease: The Interplay between Blood Cells and Soluble Factors

**DOI:** 10.3390/biom11091309

**Published:** 2021-09-04

**Authors:** Efthimia G. Pavlou, Hara T. Georgatzakou, Sotirios P. Fortis, Konstantina A. Tsante, Andreas G. Tsantes, Efrosyni G. Nomikou, Athanasia I. Kapota, Dimitrios I. Petras, Maria S. Venetikou, Effie G. Papageorgiou, Marianna H. Antonelou, Anastasios G. Kriebardis

**Affiliations:** 1Laboratory of Reliability and Quality Control in Laboratory Hematology (HemQcR), Department of Biomedical Sciences, School of Health & Caring Sciences, University of West Attica (UniWA), 12243 Egaleo, Greece; epavlou@uniwa.gr (E.G.P.); cgeorgatz@uniwa.gr (H.T.G.); sfortis@uniwa.gr (S.P.F.); bisc19678318@uniwa.gr (K.A.T.); andreas.tsantes@yahoo.com (A.G.T.); efipapag@uniwa.gr (E.G.P.); 2Blood Bank and Hemophilia Unit, Hippokration Hospital, 11527 Athens, Greece; efrosyni.nomikou@gmail.com; 3Nephrology Department, Hippokration Hospital, 11527 Athens, Greece; nancy85kap@yahoo.gr (A.I.K.); petrasdim@hotmail.com (D.I.P.); 4Laboratory of Anatomy-Pathological Anatomy & Physiology Nutrition, Department of Biomedical Sciences, School of Health & Caring Sciences, University of West Attica (UniWA), 12243 Egaleo, Greece; mvenet@uniwa.gr; 5Department of Biology, Section of Cell Biology & Biophysics, School of Science, National and Kapodistrian University of Athens (NKUA), 15784 Athens, Greece; manton@biol.uoa.gr

**Keywords:** coagulation factors, chronic kidney disease, hemodialysis, ADAMTS-13 inhibitor, platelet dysfunction, thrombotic risk, red blood cells

## Abstract

Coagulation abnormalities in renal pathology are associated with a high thrombotic and hemorrhagic risk. This study aims to investigate the hemostatic abnormalities that are related to the interaction between soluble coagulation factors and blood cells, and the effects of hemodialysis (HD) on it, in end stage renal disease (ESRD) patients. Thirty-two ESRD patients under HD treatment and fifteen healthy controls were included in the study. Whole blood samples from the healthy and ESRD subjects were collected before and after the HD session. Evaluation of coagulation included primary and secondary hemostasis screening tests, proteins of coagulation, fibrinolytic and inhibitory system, and ADAMTS-13 activity. Phosphatidylserine (PS) exposure and intracellular reactive oxygen species (iROS) levels were also examined in red blood cells and platelets, in addition to the platelet activation marker CD62P. Platelet function analysis showed pathological values in ESRD patients despite the increased levels of activation markers (PS, CD62P, iROS). Activities of most coagulation, fibrinolytic, and inhibitory system proteins were within the normal range, but HD triggered an increase in half of them. Additionally, the increased baseline levels of ADAMTS-13 inhibitor were further augmented by the dialysis session. Finally, pathological levels of PS and iROS were measured in red blood cells in close correlation with variations in several coagulation factors and platelet characteristics. This study provides evidence for a complex coagulation phenotype in ESRD. Signs of increased bleeding risk coexisted with prothrombotic features of soluble factors and blood cells in a general hyperfibrinolytic state. Hemodialysis seems to augment the prothrombotic potential, while the persisted platelet dysfunction might counteract the increased predisposition to thrombotic events post-dialysis. The interaction of red blood cells with platelets, the thrombus, the endothelium, the soluble components of the coagulation pathways, and the contribution of extracellular vesicles on hemostasis as well as the identification of the unknown origin ADAMTS-13 inhibitor deserve further investigation in uremia.

## 1. Introduction

Along with anemia, coagulation disorders constitute major hematological abnormalities observed in renal pathology [1]. Uremic toxin accumulation in patients with end stage renal disease (ESRD) promotes platelet abnormalities which are considered to be responsible for the bleeding risk that most of these patients must face [2]. However, there are many additional risk factors potentially leading to thrombotic events in this group of patients, including blood abnormalities, inflammation, comorbidities, and endothelial dysfunction [3].

At the same time, even though hemodialysis is the main renal replacement therapy used for the elimination of toxic byproducts, it is also supposed to trigger venous thromboembolism [4]. So, the existing studies in hemostatic abnormalities in ESRD are complicated by reporting increased risk for bleeding along with ongoing thrombotic events.

The present study is the first one that investigates the crosstalk between soluble coagulation factors and blood cells, its association with the hemostatic abnormalities, and the effects of hemodialysis therapy on it in ESRD patients.

## 2. Materials and Methods

### 2.1. Subjects

Thirty-two ESRD patients under maintenance hemodialysis and fifteen age- and gender-matched healthy subjects exhibiting normal hematological profile and taking no medication or food supplements were included in the study. ESRD patients have been on regular hemodialysis thrice per week (mean time on dialysis 41.3 ± 21.4 months), using high-flux biocompatible dialyzer membranes (Deerfield, USA). They were receiving rhEPO (Epoetin Alfa HEXAL, Sandoz GmbH, Kundl, Austria) (on average 6833.3 ± 3588.7 IU/week and heparin 4470 ± 1243 IU/session) and were clinically stable at the time of investigation. The primary cause of renal failure was hypertensive nephropathy (*n* = 11), polycystic kidney disease (*n* = 5), glomerulonephritis (*n* = 2), and chronic renal failure of unknown etiology (*n* = 14). Patients with diabetes mellitus, uncontrolled hypertension, active infections, malignant, inflammatory, autoimmune or hematological diseases, or those who needed to get a blood transfusion over the past 3 months were excluded from the study. All ESRD patients and control subjects were negative for anti-b2-glycoprotein I (b2-GPI) and anti-cardiolipin antibodies as well as for lupus anticoagulants. Activated Protein C Resistance assay had been also performed to exclude Factor V Leiden deficiency in all participants.

Whole blood samples from healthy subjects and ESRD patients, before and immediately after the end of HD session, were collected in ethylenediaminetetraacetic acid (EDTA) and 3.2% sodium citrate blood collection tubes (BD Vacutainer Blood Collection Tubes, BD Biosciences, San Jose, CA, USA). The study has been submitted and approved by the Research Ethics Committee of the Department of Biomedical Sciences/UNIWA. Investigations were carried out in accordance with the principles of the Declaration of Helsinki. Written informed consent was obtained from all participants.

### 2.2. Material Supplies

All materials and common chemicals were obtained from Sigma-Aldrich (Munich, Germany), unless otherwise stated. Detection of lupus anticoagulants was performed by using qualitative HemosIL dRVVT Screen/dRVVT Confirm assays in ACL TOP Hemostasis Analyzer (Instrumentation Laboratory, Bedford, MA, USA). For the quantitative determination of anti-b2 glycoprotein and anti-cardiolipin IgG/IgM, Elisa kits by Delta Biologicals Srl were used according to manufacturer instructions (Erba Diagnostics Inc., Pomezia, Roma, Italy). Assessment of Activated Protein C Resistance was performed by STA-STACLOT APC-R (Diagnostica Stago, Parsippany, NJ, USA). Antibodies used for flow cytometry were obtained from BD Biosciences (San Jose, CA, USA).

### 2.3. Hematological and Serum Biochemical Analysis

General blood tests were performed by using a Siemens Advia 2120i Hematology Analyzer. For the biochemical analysis of serum components (urea, creatinine, uric acid, parathormone, glucose, cholesterol, triglycerides, calcium, phosphorus, potassium, sodium, chlorine, magnesium, iron, ferritin, total iron-binding capacity, proteins, albumin, b2-microglobulin, serum glutamyl oxalate transaminase, serum glutamyl pyruvate transaminase, gamma-glutamyl transferase, alkaline phosphatase, total bilirubin, indirect bilirubin, direct bilirubin, creatine phosphokinase, amylase, lactate dehydrogenase, and vitamin-D) the automatic Clinical Chemistry Analyzer ARCHITECT C16000 (Abbott, Illinois, IL, USA) was used. hs-CRP levels were determined with a commercial kit [Abbott Laboratories, Hellas (Greece)] in the Architect C8000 analyzer. Plasma free hemoglobin (fHb) was calculated by the method of Harboe [5].

### 2.4. Measurement of Intracellular ROS Levels

Intracellular levels of reactive oxygen species (ROS) were detected by using the membrane-permeable fluorescent probe CMH_2_DCFDA (Invitrogen, Molecular Probes, Eugene, OR, USA). Briefly, red blood cells (RBCs) diluted at 0.4% hematocrit in PBS/5 mM D-glucose were loaded with 20 μM CM-H_2_DCFDA for 60 min at 37 °C, in the dark. For the determination of the intracellular ROS levels in platelets (PLTs), platelet-rich plasma (PRP) was obtained after centrifugation at 200× *g* for 20 min, at 24 °C. PRP was then diluted in HEPES Tyrode’s buffer (138 mM NaCl, 2.6 mM KCl, 5.5 mM glucose, 5 mM Hepes/NaOH, 0.49 mM MgCl_2_, 0.36 mM NaH_2_PO_4_H_2_O, 12 mM NaHCO_3_) containing 0.1% BSA and incubated with 10 μM CM- H_2_DCFDA for 30 min at 37 °C in the dark. Mean fluorescence index and percentage of ROS-positive cells were determined by flow cytometry in a FACSCanto II Cytometer (BD Biosciences, San Jose, CA, USA). Data analysis was performed using the BD FACSDiva™ Software. A minimum of 30,000 events was acquired for RBCs and 100,000 events for platelets on each sample.

### 2.5. Measurement of Intracellular Free Calcium Levels ([iCa^2+^])

Intracellular calcium levels were detected by using the membrane-permeable fluorescent probe Fluo-4, AM (Invitrogen, Molecular Probes, Eugene, OR, USA). Briefly, RBCs diluted in a buffer containing 145 mmol/L NaCl, 7.5 mmol/L KCl, 10 mmol/L Hepes/NaOH, 1.8 mmol/L CaCl_2_, 10 mmol/L glucose and 10 mmol/L sodium pyruvate (pH 7.4) were loaded with 1 μM reagent for 50 min at 37 °C in the dark. For the determination of the intracellular calcium levels in PLTs, PRP was diluted in HEPES Tyrode’s buffer before incubation with 1 μM Fluo-4 AM for 30 min at 37 °C, in the dark. Mean fluorescence index of red blood cells and platelets was determined by flow cytometry, as mentioned above.

### 2.6. P-Selectin Expression and Phospatidylserine Exposure

Phenotyping of RBCs was performed by multicolor flow cytometry using phycoerythrin (PE)-Annexin V apoptosis kit and fluorescein isothiocyanate (FITC)-conjugated anti-CD235 (BD Biosciences, San Jose, CA, USA), as previously described [6].

For the identification of platelet activation in PRP surface expression of P-selectin (CD62P) and phosphatidylserine (PS) were measured. Briefly, PRP diluted in Annexin V-binding buffer (Hepes/NaOH 0.01 M, NaCl 140 mM, CaCl_2_ 2.5 mM) was incubated for 15 min at room temperature in the dark with allophycocyanin (APC)-conjugated anti-CD62P and PE-Annexin V apoptosis kit along with the platelet gating marker PE Cy5-CD41a (BD Biosciences, San Jose, CA, USA). Reactions were stopped with Annexin V-binding buffer and measurements were performed by flow cytometry.

### 2.7. Platelet Function Assays

Platelet adhesion and aggregation functions were evaluated by using the Dade PFA Collagen/EPI Test Cartridge and Dade PFA Collagen/ADP Test Cartridges, according to the manufacturer’s instructions (Siemens Healthcare Diagnostic, Erlangen, Germany), in the PFA-100 Analyzer (SIEMENS, Malvern, PA, USA).

### 2.8. Screening Test and Coagulation Factors Analysis

Prothrombin time-international normalized ratio (PT-INR) was measured with Thromborel S Reagent. Activated Partial Thromboplastin Time (aPTT) was measured with Pathromtin SL Reagent (Siemens Healthcare Diagnostic, Erlangen, Germany). The turbidimetric measurement of von Willebrand activity (Ristocetin Cofactor assay, RiCof) was performed with Innovance VWF Ac kit (Siemens, Marburg, Germany). Berichrom Factor XIII and Berichrom PAI assays were used for chromogenic determination of Factor XIII and plasminogen activator inhibitor’s activity, respectively (Siemens Healthineers, Erlangen, Germany). Quantitative determination of fibrinogen was performed by Multifibren U kit and cross-linked fibrin degradation products by Innovance D-dimer kit (Siemens Healthineers, Erlangen, Germany), according to the manufacturer’s recommendations. All the above-mentioned parameters were measured in a blood Coagulation Analyzer BCS XP System (Siemens Healthineers, Erlangen, Germany).

STA-Deficient VIII, STA-Deficient IX, and STA-Immunodef XI kits were used for measurement of factors’ VIII, IX, and XI activity, respectively, along with the STA-PTT A kit, whereas STA Immunodef XII and C.K. Prest were used for the determination of FXII activity. Activities of factors II, V, VII, and X were measured using STA-Deficient II, STA-Deficient V, STA-Deficient VII, and STA-Deficient X kits, respectively, with the NeoPTimal reagent (Diagnostica Stago, Parsippany, NJ, USA). Calcium chloride solution 0.025 mol/L was also used for all the activity measurements (Diagnostica Stago, Parsippany, NJ, USA). Antigenic assay of free Protein S was performed by the STA-Liatest Free Protein S kit. Von Willebrand factor antigen was measured immuno-turbidimetrically by using the STA-Liatest VWF:Ag kit, according to the manufacturer’s recommendations (Diagnostica Stago, Parsippany, NJ, USA). Quantitative determination of the plasminogen and Protein C activities was performed calorimetrically by using the STA-Stachrom Plasminogen and Protein C reagents kit. Those measurements were performed in the STAR STAGO hemostasis analyzer.

Protein S activity was determined with HemosIL Protein S activity kit by measuring clotting time (Instrumentations Laboratory, Bedford, MA, USA) as per manufacturer’s instructions. A chromogenic assay was used for the determination of ATIII activity using the Liquid Antithrombin kit by Hemosil (Instrumentations Laboratory, Bedford, MA, USA). An ACL TOP Hemostasis Analyzer (Instrumentations Laboratory, Bedford, MA, USA) was used for the measurement of the above-mentioned parameters.

### 2.9. ADAMTS Activity and Inhibitor Test

Quantitative measurement of ADAMTS-13 activity and inhibitor tests were performed by using LIFECODES ATS-13 activity assay (Immucor GTI Diagnostics, Waukesha, WI, USA), according to the manufacturer’s instructions. Functional inhibitor assays were performed following mixing with normal plasma. The ATS-13 activity assay (that is based on the fluorescence resonance energy transfer technology-FRET) was performed in the Thermoscientific Fluoroskan microplate reader (Waltham, MA, USA).

### 2.10. Statistical and Network Analysis

All experiments were performed twice, unless otherwise stated. The Statistical Package for Social Sciences (IBM SPSS Software; version 26.0 for Windows IBM Corp., Armonk, NY, USA; administrated by UNIWA) was used for statistical analysis. All variables were tested for normal distribution profile. Between group differences in normally distributed parameters were evaluated by *t*-test using a Bonferroni correction (or Mann–Whitney U test for not normally distributed parameters). Paired samples were assessed by *t*-test or two-related samples test, for normally or not normally distributed parameters, respectively, for the consideration of hemodialysis effect (pre- versus post-hemodialysis parameters). Pearson’s and Spearman’s tests were performed to assess the correlation factor (r) between measurements following normal or not normal distribution profiles, respectively. Significance was accepted at *p* < 0.05. Significant correlations of serum, plasma, PLT, and RBC variables were topologically represented in undirected biological networks by using the Cytoscape version 3.8.2 application [7]. The length of each edge was inversely proportional to the r value (namely, the shorter the edge length, the higher the r value between the two interrelated nodes was).

## 3. Results

### 3.1. Baseline Hematological and Serum Biochemical Profile

The main demographic, therapy-associated, and hematological characteristics are presented in Table 1. The hematological profile of ESRD patients showed the anticipated anemia markers, namely, low levels of RBC count (RBCs), hematocrit (Hct), hemoglobin concentration (Hb), and mean corpuscular Hb concentration (MCHC) (Table 1). Hematocrit levels did not differ significantly among patients. Despite being within normal range, patients had statistically significant higher neutrophils and monocytes but lower lymphocytes percentages compared to the co-studied cohort of healthy subjects. Mean platelet volume (MPV) was also higher than control.

Accumulation of uremic toxins (urea, creatinine, parathormone-PTH, and b2-microglobulin) went in parallel with the typical inflammatory profile of ESRD patients, as directly manifested by the pathologically increased ferritin and hsCRP levels, and indirectly through the aberrant neutrophils/lymphocytes ratio (Table 1). Serum cations including calcium, phosphorous, potassium, sodium, and magnesium were also significantly increased compared to the co-studied cohort of healthy subjects, though not at pathological levels (Table 1). Post-hemodialysis there was an overall improvement in the uremic indexes, though creatinine remained pathologically high (Table 1). The values of the other parameters were adjusted after dialysis session according to the fluid loss (data not shown).

### 3.2. Functional Analysis of Platelets (PFA/EPI, PFA/ADP)

Platelet functionality towards primary hemostasis was then assessed following addition of the agonists ADP and epinephrine (EPI) in PLTs laid on collagen-coated (Col) cartridge membranes. In both assays the closure times were substantially prolonged in uremic patients (Col/EPI and Col/ADP, Figure 1), both before (206.7 ± 58.8 and 169.3 ± 77.3, respectively) and soon after (215.1 ± 70.6 and 154.5 ± 53.2 s, respectively) the HD session compared to the healthy control cohort of the study (102.8 ± 17.2 and 88.4 ± 21.6, respectively) and the normal range (Col/EPI < 142 s, Col/ADP < 102 s), revealing a clear defect of ESRD PLTs towards hemostasis. HD treatment did not affect any of those measurements (Figure 1).

### 3.3. PLT Activation Signaling and Oxidative Stress

Expression of P-selectin (CD62P), a marker of platelet activation, was higher in ESRD PLTs compared to controls before (21.6 ± 15.8 vs. 6.8 ± 3.5%) and after (15.1 ± 10.3%) HD (Figure 2a). Moreover, an increased percentage of the activated PLTs exhibited surface expression of phosphatidylserine (PS), independently of the HD session (Figure 2b). Hemodialysis decreased PLT activation but had no effect on PS exposure on them (Figure 2b,c). Intracellular ROS accumulation was also higher in uremic PLTs compared to healthy control (1.91 ± 0.68 vs. 0.91 ± 0.44 %, respectively). Dialysis treatment leads to a significant decrease in the percentage of ROS+ platelets, though they remain at pathologically high levels post-HD (1.42 ± 0.76%) (Figure 2a). In contrast to ROS accumulation, the intracellular calcium levels in resting ESRD PLTs before (1854 ± 476 MFI) and after (1883 ± 456 MFI) the HD session were similar to healthy PLTs (2001 ± 182 MFI).

### 3.4. Stress Signaling in Red Blood Cells (RBCs)

We then sought to explore whether the RBCs in ESRD (that contribute by many pathways to the physiological coagulation [8]) were characterized by similar stress and activation signals with those detected in PLTs. Indeed, we found an increased percentage of ESRD RBCs exposing PS compared to controls (1.60 ± 0.93% vs. 0.69 ± 0.30%) before and after (1.60 ± 1.09%) HD (Figure 3b). In similarity with the ESRD PLTs (Figure 2b), the HD did not affect the PS surface exposure on RBCs, at least shortly after the session. In addition to that, the vast majority of the ESRD RBCs were ROS^+^ (92.3 ± 5.3%), again, at higher levels compared to control RBCs (79.5 ± 7.9%) (Figure 3a). In contrast to the ESRD PLTs (Figure 2a), the HD had no effect on the percentage of ROS^+^ RBCs. Finally, control levels of calcium accumulation were detected in ESRD RBCs before (792 ± 137 MFI) and after HD (810 ± 137 vs. 905 ± 213 MFI in control). In that case, however, the HD exerted a slight triggering effect (*p* < 0.05 pre- vs. post-HD).

### 3.5. Coagulation (Secondary Hemostasis) Screening Test

As reflected in the average fluctuations shown in Table 2, the aPTT and the PT-INR were pathologically increased in 65% and 24% of the ESRD patients, respectively. In most of the sessions, hemodialysis had an ameliorative effect on both parameters (Table 2). Fibrinogen and D-dimers concentrations were pathologically high in 75 and 60% of the patients, respectively, but in contrast to aPTT and PT-INR, HD had either no effect (D-dimers) or an aggravating effect (fibrinogen) on their levels (Table 2).

### 3.6. Functional Analysis of Factors Involved in the Coagulation Cascade and in Fibrinolysis

With the exceptions of factors VIII, XIII, and vW, the activities of coagulation factors were within the normal range in ESRD patients. In fact, the FII, FV, and FXII exhibited lower activity compared to the cohort of healthy subjects. Pathologically increased activities were measured for factors VIII, XIII, vW, and Ricof in most of the ESRD patients (eg. in 65% of them for vWF) before HD. Hemodialysis enhanced the activities of several factors (V, VIII, XII, RiCof, and vW) leading to pathological post-HD levels in many events (>80% for VIII, Ricof, and vW). In accordance with the increased activity of vWF, all the ESRD patients exhibited significantly low ADAMTS-13 and high ADAMTS-13 inhibitor activities before and after HD. Of note, HD further triggered the activity of ADAMTS-13 inhibitor (Table 2).

Regarding the fibrinolytic system, control levels of both PAI activity and plasminogen concentration were measured before and after the HD session, being, however, higher than the average value of healthy cohort. HD seemed to exert a slight triggering effect on plasminogen concentration. In the same context, the activities of the physiological inhibitors of coagulation antithrombin III (ATIII) and Protein C were within the normal range, despite enhancement by the HD session. The activity ranges of Protein S and plasma-free Protein S antigen were also within control levels, though the activity of the plasma free Protein S antigen was lower compared to that of healthy cohort, and further decreased by HD (Table 2).

### 3.7. Biological Networks

Biological networks were constructed for the visualization of statistically significant correlations among uremic toxins, coagulation factors, and blood cell characteristics (Figure 4). As expected, there were strong correlations between variations in uremic toxins and coagulation parameters. For example, urea and creatinine were reversibly related to ATIII pre-dialysis (Figure 4, light-green square). Post-dialysis, the levels of these toxins seemed to correlate positively with the decreased phosphatidylserine exposure in RBCs but inversely with II and V (Figure 4, light-orange rectangle). ROS concentration in PLTs post-dialysis was positively correlated with a variety of factors involved in hemostatic pathways such as fibrinogen (r = 0.579, *p* = 0.019), PT/INR (r = 0.525, *p* = 0.037), D-dimers (r = 0.526, *p* = 0.036), and PS/CD62P^+^ PLTs (r = 0.548, *p* = 0.043). In the same direction, ROS^+^ RBCs showed strong connections with several coagulation factors, including RiCof (r = 0.539, *p* = 0.017), FVIII (r = 0.526, *p* = 0.021), and FXIII (r = 0.561, *p* = 0.012) (Figure 4, yellow rectangle). Finally, before dialysis session percentage of PS^+^ or PS^+^/CD62P^+^ PLTs showed positive correlations with ATIII (r = 0.594, *p* = 0.032), iCa^2+^ PLTs (r = 0.586, *p* = 0.028), FX (r = 0.790, *p* = 0.007), and plasminogen (r = 0.835, *p* = 0.001) (Figure 4, light-blue square).

## 4. Discussion

Uremic toxin accumulation in end stage renal disease is associated with anemia, inflammation, and coagulation disorders. Hemodialysis, the standard symptomatic therapy in ESRD, generally improves anemia but is not able to entirely relieve inflammation, whereas there are discrepancies in the field regarding its effect on the coagulation status. Despite the established contribution of RBCs to the coagulation cascade, their possible relation with platelet dysfunctions and the disturbed coagulation pathways reported in ESRD patients on HD have not been clarified so far.

Our patient cohort was representative of the disease, as characterized by anemia and inflammation signs, including hsCRP, serum ferritin, neutrophils/lymphocytes ratio [9,10], and fibrinogen, an acute phase reactant. As expected, positive co-variations were observed among those parameters (e.g., between fibrinogen and hsCRP (r = 0.469, *p* = 0.028)). Of note ferritin covaried with the anemia levels as it was found inversely correlated with the Hb concentration, the Hct, and the RBC count (Figure 4) [11].

In accordance with previous studies [1,12], the functional abnormalities of uremic platelets are reflected in the prolonged closure times observed after addition of the agonists, ADP, and epinephrine. There is evidence for a causal linkage with the uremic toxins [13], not supported, however, by other studies [14]. Despite the expected ameliorating effect of HD on platelet function [15,16], we did not observe any improvement post-HD in our patients who presented normal or increased vWF activity; thus ruling out the existence of von Willebrand disease. These data suggested a complex pathophysiological basis of platelet abnormalities in ESRD, probably involving intrinsic cell disturbances, [17] like impaired binding to vessel wall and fibrinogen, defective metabolism, decreased content of granules, and impaired intracellular signaling among others [18]. Moreover, the prolonged Col/EPI closure time has been previously related to the low Hct levels [19].

The prolonged activated Partial Thromboplastin Time (aPTT) confirmed the increased bleeding risk in our patient’s pre-dialysis, as suggested by others [1]. In combination with the high PT/INR ratio, the whole picture included abnormalities in both intrinsic and extrinsic coagulation cascades. Normalization of PT/INR and aPPT soon after the HD session might be partially attributed to the concurrent increase in the activity of coagulation factors and natural inhibitors. This is a probable result of removing excess of water and dialyzable components. As shown in the interactomes of Figure 4, both urea and creatinine varied inversely with the ATIII levels.

Pathological levels of FVIII is another common disturbance in ESRD patients [1,20,21]. In our cohort, FVIII variation followed that of vWF antigen and activity pre-HD. vWF has a dual role in hemostasis, as both a carrier of FVIII and as a bridging molecule for intercellular interactions between platelets or between platelets and sub-endothelium. The increased vWF activity obviously did not compensate for the prolonged PFA, confirming our hypothesis about endogenous cell disturbances in ESRD. In contrast to Casonato et al. [20], but in agreement with Holden et al. [22], hemodialysis significantly increased factors VIII, Ricof, and von Willebrand in our cohort of patients, suggesting a negative effect on endothelium integrity and probably on thrombotic risk [23,24,25].

Increased activity of vWF could be attributed to the accumulation of inflammatory cytokines [26] or to the sharp decline in ADAMTS-13 activity (<5% of normal values in our patients). ADAMTS-13 is a proteolytic enzyme responsible for the removal of high molecular weight vWF multimers from circulation. The inverse covariation of vWF with ADAMTS-13 has been considered a prothrombotic state in chronic kidney disease [27]. While many potential inhibitors of ADAMTS-13 activity have been proposed (e.g., autoantibodies [28], plasma IL-6 [29], human neutrophil peptides [30], thrombospondin-1 [31], cell-free plasma Hb, platelet factor 4 (PF4) [10]), none of them has been tested in ESRD, highlighting a significant knowledge gap in the field of hemostasis in uremia.

Abnormalities in fibrinogen levels lead either to hyper- or hypofibrinogenemia. Both of them carry a prothrombotic potential probably due to intrinsic fibrinogen disorders [32,33,34]. These disorders could also affect fibrinolysis, a process which includes activation of tissue plasminogen activator (t-PA) factor and regulation by Plasminogen Activator Inhibitor (PAI) [35,36]. In our patient cohort, a mixed biochemical phenotype regarding fibrinolysis was observed, which includes hyperfibrinogenemia [21] as a probable result of the inflammatory status [37] predisposing to thrombosis [38] and at the same time, pathologically increased D-dimers suggesting increased plasmin activity and hyperfibrinolysis. It was expected that fibrinogen positively covaried with both the plasminogen and D-dimers levels. Expression of P-selectin on activated platelets also predisposes to thrombosis by promoting aggregation [39], thrombus formation at site of endothelium damage, and increased incorporation of leukocytes and fibrin into the thrombus [40]. Ultra large multimers of vWF [41] and PS externalization could also contribute to these prothrombotic procedures. The surface expression of this aminophospholipid in activated platelets promotes the assembly of prothrombinase complex in the presence of FV, FΧ, and calcium ions [42]. PS is also a powerful apoptotic marker, linked to an iCa^2+^-dependent increase in caspase-3 activation [43]. In our study, the positive correlation of PS^+^ or PS ^+^/CD62P^+^ PLTs with intracellular calcium and FX or FXI levels (Figure 4) probably signify the proapoptotic and prothrombotic dynamics of those platelets. This prothrombotic potential could probably be attenuated to some extent by an increase in proteins of fibrinolytic or inhibitory system of hemostasis, as it is indicated by their positive correlation with ATIII and plasminogen levels.

PLT activation might happen during the dialysis session [44]; however, dialysis often reduces the PLT activation markers [44,45]. This was also observed in our patients through decrease in the CD62P^+^ PLTs, as a probable result of HD-induced cell clearance [46] or increased microvesiculation [47]. The persistently increased PFA values of ESRD patients that were not affected by the dialysis session in contrast to the P-selectin expression levels that were decreased, imply that platelet dysfunction (reflected in the PFA values) is independent of P-selectin expression in ESRD patients, as previously described in children with iron deficiency anemia [48]. The significant increase in fibrinogen and vWF levels, two components of platelet granules that are released following PLT stimulation, also implies an ongoing PLT activation and degranulation during HD [49]. The HD-induced PLT degranulation may also be contributing to ADAMTS inhibition through the release of PF4 and thrombospondin-1, as previously suggested [10,31].

The increased ROS levels in platelets may be associated with the activation of collagen receptor [50] and the toxicity of uremic factors [51] since they effectively decreased post-dialysis. As an additional marker of cellular activation, the ROS^+^ PLTs can regulate thrombus formation by inducing fibrinogen adhesion and granule release [52]. Actually, the probable hemostatic role of ROS^+^ PLTs is depicted in their significant correlations with fibrinogen, D-dimers, and INR values (Figure 4). Of note, increased generation of intracellular ROS has been previously associated with P-selectin expression [53], as currently observed in our patient cohort (Figure 4, r = 0.548, *p* = 0.043).

Finally, according to our results, RBCs might represent another piece in the complex coagulation puzzle of ESRD. Increased intracellular ROS and PS exposure levels have been previously reported in the context of RBC pathology in anemic renal disease patients [54]. Nowadays, RBCs are considered dynamic regulators of the coagulation mechanisms, through NO scavenging, ADP and thromboxane A_2_ release following hemolysis, membrane exovesiculation, and direct interaction with platelets through Fas-FasL linkages [8,55,56]. According to the biological networking analysis, the ROS^+^ RBC percentage or the intracellular ROS levels positively covaried with the activities of several coagulation factors, including vWF:Ag, Ricof, FVIII, and FXIII and negatively with the PFA/EPI time post-dialysis, suggesting a similar functional role in ESRD. Actually, there is evidence for a dynamic interaction between FXIII, vWF, and RBCs during thrombus formation [57]. It is of interest that similar connections were detected with the ROS^+^ PLTs pre-dialysis but not following it, probably due to the selective beneficial effect of HD on PLT (but not on RBC) ROS burden. Apart from these, RBCs subjected to oxidative stress could also contribute to the procoagulant status of ESRD patients through the release of PS^+^ microvesicles [58,59]. The significantly increased fibrinogen observed post-HD in ESRD patients might further interact with RBCs promoting RBC-RBC and RBC-PLT aggregates [60,61].

## 5. Conclusions

This study provides evidence for a complex coagulation phenotype in ESRD that is configurated by soluble and corpuscular factors, their interactions, and the effect of dialysis therapy on both. Signs of increased bleeding risk at the pre-dialysis period, like the long aPTT, PT/INR, and PFA times, coexisted with soluble (e.g., fibrinogen, vWF/ADAMTS activity) and cellular (eg. activated PS^+^RBCs, and PLTs) prothrombotic features in a general hyperfibrinolytic state (DD). Hemodialysis seems to augment the prothrombotic potential through increases in the activity of several coagulation factors, fibrinogen, and ADAMTS-13 inhibitor; however, it mitigates in part the activation state of platelets and normalizes the aPTT and PT/INR times. Τhe persisted PLT dysfunction might counteract the increased predisposition to thrombotic events post-dialysis. RBCs abnormalities seem to participate in the prothrombotic potential of uremic blood, with no detected alleviating effect by the HD side. The identification of the unknown origin of ADAMTS-13 inhibitor may probably be the lost piece of the puzzle that will unveil the full picture of hemostasis in uremia. Apart from this, the possible interaction of activated or stressed RBCs with PLTs, the thrombus, the endothelium, and the soluble components of the coagulation pathways, as well as the additional contribution of extracellular vesicles on these procedures deserve further investigation for the elucidation of coagulation state and its variation in ESRD patients.

## Figures and Tables

**Figure 1 biomolecules-11-01309-f001:**
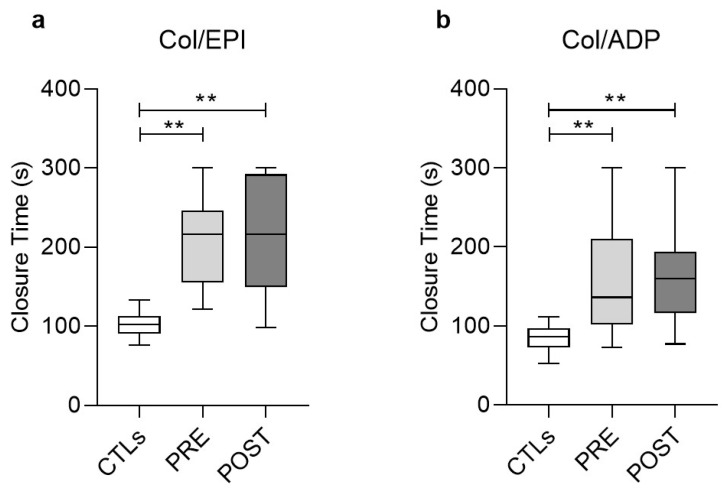
Boxplots showing the closure time measured in the Platelet Function Analysis of ESRD patients, pre-HD (PRE) and post-HD (POST), compared to healthy subjects (CTLs), after the addition of agonists. (**a**) Collagen with epinephrine, (**b**) Collagen with ADP. (**) *p* < 0.01 vs. controls. Col, Collagen; EPI, epinephrine; ADP, adenosine diphosphate.

**Figure 2 biomolecules-11-01309-f002:**
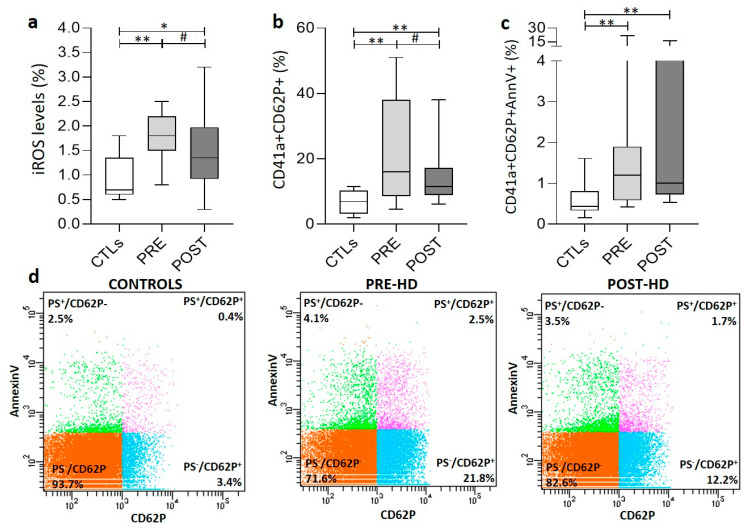
Variation in platelet characteristics in ESRD patients and healthy subjects (CTLs). (**a**) Percentage of ROS+ platelets (iROS). (**b**) Percentage of CD62P+-platelets. (**c**) Percentage of CD62P+/AnnV+ platelets. (**d**) Representative FACS plots of AnnV-PE and CD62P-APC labeled platelets from ESRD patients, before and after dialysis session, and healthy subjects. PRE, POST: ESRD patients pre-HD and post-HD, respectively (*) *p* < 0.05 vs. controls (**) *p* < 0.01 vs. controls, (#) *p* < 0.01 pre- vs. post-HD.

**Figure 3 biomolecules-11-01309-f003:**
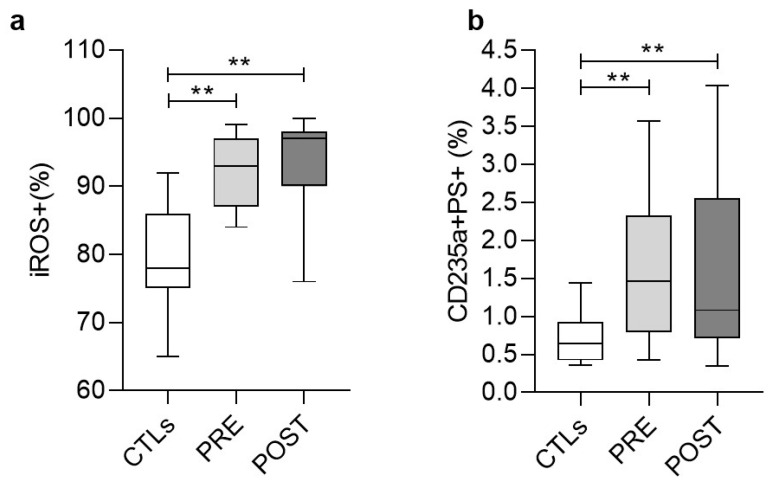
Variation in the percentages of ROS+ (**a**) and PS+ (**b**) RBCs in ESRD patients and healthy subjects (CTLs) measured by flow cytometry. PRE, POST: ESRD patients pre-HD and post-HD, respectively (**) *p* < 0.01 vs. controls.

**Figure 4 biomolecules-11-01309-f004:**
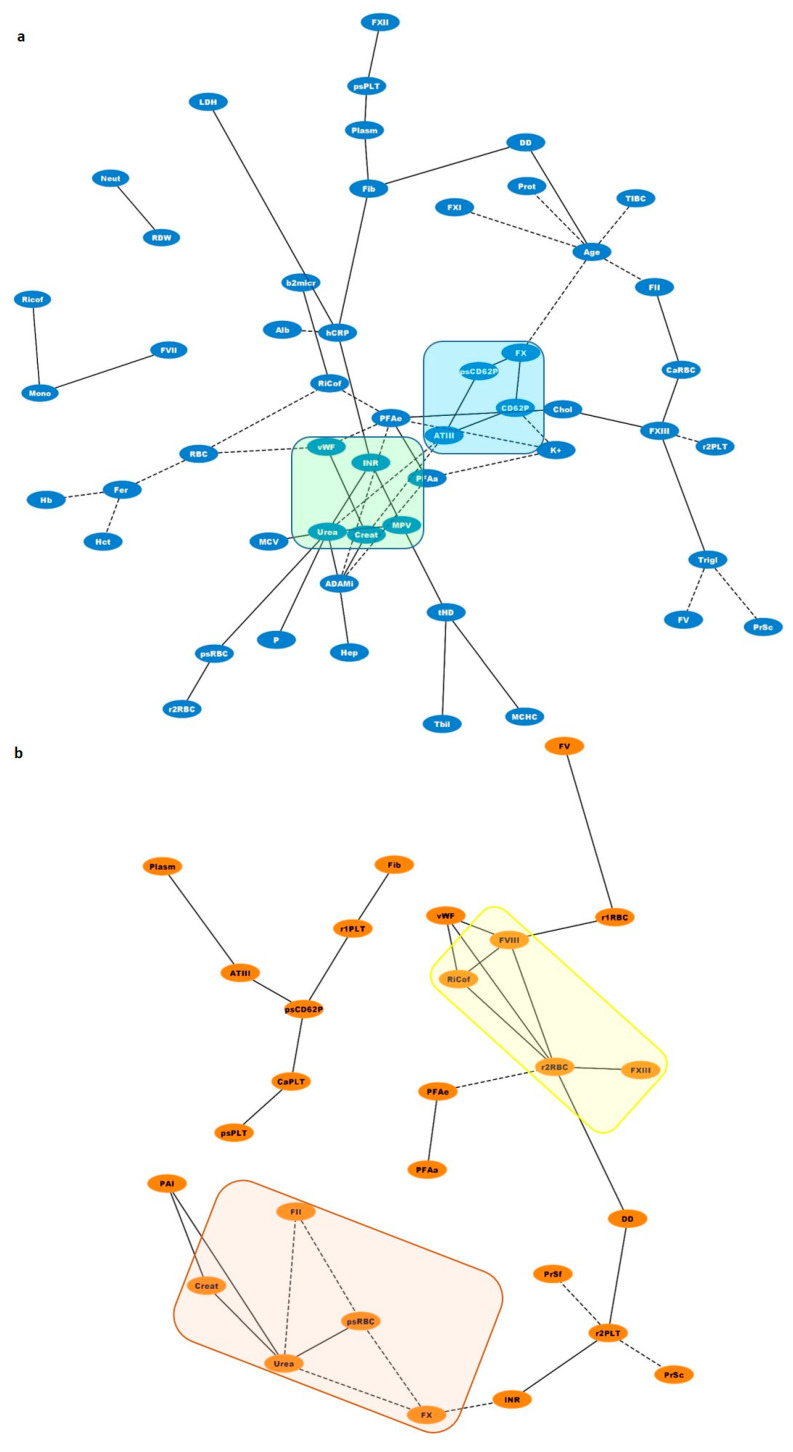
Network analysis of statistically significant (*p* < 0.05) positive (black lines) and negative (dotted black lines) correlations between soluble coagulation factors and blood cell features in end stage renal disease (**a**) before (blue nodes) and (**b**) soon after (orange nodes) the dialysis session. Squares and rectangles highlight correlations of outmost importance discussed in Section 3.7. The shortest the edge length, the higher the r value between the two interrelated nodes was. Abbreviations: Age, age of patients (years); ADAMi, ADAMTS-13 inhibitor; Alb, albumin; ATIII, antithrombin III; b2micr, b2-microglobulin; CaPLT, platelet intracellular calcium (MFI); CaRBC, intracellular calcium RBCs (MFI), Chol, Cholesterol; Creat, creatinine, CD62P, CD62P^+^ PLTs (%); DD, D-dimers; Fer, Ferritin; Fib, Fibrinogen; Hb, Hemoglobin; Hct, Hematocrit; hCRP, highsensitivity C-Reactive Protein; Hep, Heparin dose; INR, Prothrombin time—International Normalized Ratio; K^+^, serum potassium; LDH, lactate dehydrogenase; MCHC, mean corpuscular hemoglobin concentration; MCV, mean corpuscular volume; Mono, Monocytes (%); MPV, mean platelet volume (fL); Neutr, Neutrophils (%); P, serum phosphorous; PAI, Plasminogen Activator Inhibitor; PFAa, PFA Col/ADP; PFAe, PFA Col/EPI; Plasm, Plasminogen; Prot, Plasma proteins; PrSc, Protein S activity; PrSf, Protein S Antigen free; psCD62P, PS/CD62P+ PLTs (%); psPLT, PS^+^ PLTs; psRBC, PS^+^ RBS (%); RBC, RBC count; RDW, RBC distribution width; r1PLT, PLTs ROS (MFI); r1RBC, RBCs ROS (MFI); r2PLT, PLTs ROS (%); r2RBC, RBCs ROS (%); Tbil, total bilirubin; tHD, time under HD therapy (months); Trigl, Triglycerides.

**Table 1 biomolecules-11-01309-t001:** Demographic, therapy-associated, hematological, and serum biochemical characteristics of patients with ESRD and healthy subjects.

Characteristics	Patients (*n* = 32)		Controls (*n* = 15)	Normal Range
	Pre-HD	Post-HD		
HD treatment (months)	41.3 ± 21.4		-	-
Age (years)	59 ± 11.4		50 ± 9.8	-
EpO dose (IU/wk)	6833.3 ± 3588.7		-	-
Weight (kg)	82.2 ± 11.1	80.2 ± 11.8	85.6 ± 13.2	-
General blood test				
WBC (×10^3^/µL)	7.60 ± 2.70	N/D	7.10 ± 1.80	5.2–12.4
Neutrophils (%)	64.7 ± 8.3 *	N/D	59.0 ± 8.0	40.0–74.0
Lymphocytes (%)	20.7 ± 6.3 **	N/D	28.7 ± 6.6	19.0–48.0
Monocytes (%)	7.4 ± 1.6 **	N/D	6.1 ± 1.6	3.4–9.0
Eosinophils (%)	4.9 ± 3.9	N/D	2.9 ± 1.9	0.0–7.0
Basophils (%)	0.7 ± 0.4	N/D	0.8 ± 0.3	0.0–1.5
Neutrophils/Lymphocytes ratio	**3.5 ± 1.6 ****	N/D	1.9 ± 0.6	1.3–3.5
RBC (×10^6^/µL)	**3.80 ± 0.59 ****	N/D	4.60 ± 0.46	4.2–6.1
Hb (gr/dL)	**11.1 ± 1.3 ****	N/D	13.5 ± 1.6	12.0–18.0
Hct (%)	**34.3 ± 4.3 ****	N/D	40.6 ± 4.4	37.0–52.0
MCV (fL)	91.1 ± 9.7	N/D	88.7 ± 5.3	80.0–99.0
MCH (pg)	29.4 ± 3.4	N/D	30.5 ± 2.0	27.0–31.0
MCHC (gr/dL)	**32.3 ± 0.9 ***	N/D	34.4 ± 0.7	33.0–37.0
RDW (%)	15.8 ± 1.1	N/D	13.0 ± 3.0	11.5–14.5
PLT (×10^3^/µL)	240.8 ± 82.0	N/D	250.1 ± 40.6	130.0–400.0
MPV (fL)	10.0 ± 0.6 **	N/D	7.8 ± 1.3	7.2–11.1
Serum biochemical analysis				
Glucose	86.8 ± 22.2	N/D	89.5 ± 12.1	70–105
Urea (mg/dL)	**129.3 ± 27.9 ****	39.1 ± 12.7 ^#^	33.9 ± 9.4	18.0–55.0
Creatinine (mg/dL)	**10.42 ± 3.10 ****	**3.90 ± 1.42 **^#^**	0.92 ± 0.16	0.72–1.25
Uric Acid (mg/dL)	6.1 ± 1.0 **	N/D	4.4 ± 1.5	3.5–7.2
Cholesterol (mg/dL)	170 ± 42	N/D	178 ± 42	0.0–200.0
Triglycerides (mg/dL)	126.7 ± 52.8	N/D	115.2 ± 61.1	0.0–150.0
PTH (pg/mL)	**388.3 ± 216.2 ****	N/D	49.7 ± 15.5	15.0–68.0
Calcium (mg/dL)	8.5 ± 0.7 **	N/D	9.5 ± 0.2	8.4–10.2
Phosphorus (mg/dL)	4.5 ± 1.3 **	N/D	3.1 ± 0.5	2.4–4.7
Potassium (mmol/L)	4.95 ± 0.61 **	N/D	4.47 ± 0.39	3.5–5.1
Sodium (mmol/L)	136.6 ± 2.3 **	N/D	139.3 ± 2.3	136.0–145.0
Chlorine(mmol/L)	102.2 ± 2.9	N/D	102.1 ± 2.8	98.0–107.0
Magnesium (mg/dL)	2.28 ± 0.37 *	N/D	2.02 ± 0.02	1.60–2.60
Iron (mg/dL)	69.2 ± 25.0	N/D	88.0 ± 39.8	50–150 (females); 60–160 (males)
Ferritin (ng/mL)	**263.3 ± 153.3 ****	N/D	53.8 ± 29.1	14.0–233.0 (females); 16.4–293.3 (males)
TIBC (mg/dL)	251.6 ± 53.3 **	N/D	342.0 ± 50.7	225.0–480.0
B12 (pg/mL)	602.6 ± 265.5	N/D	414.3 ± 179.5	179.0–1162.0
Folate (ng/mL)	14.60 ± 12.50	N/D	5.30 ± 2.60	2.5–17.0
Proteins (mg/dL)	6.90 ± 0.60	N/D	6.80 ± 0.60	6.40–8.30
Albumin (gr/dL)	4.05 ± 0.50 **	N/D	4.47 ± 0.33	3.50–5.00
b2-microglobulin (mg/L)	**33.3 ± 11.4 ****	N/D	1.4 ± 0.5	0.71.8
SGOT (U/L)	11.4 ± 5.0 **	N/D	18.1 ± 5.9	5.0–34.0
SGPT (U/L)	12.00 ± 4.06 **	N/D	22.00 ± 15.00	0.0–55.0
γGT (U/L)	25.4 ± 15.3	N/D	17.9 ± 8.0	12.0–64.0
ALP (U/L)	89. ± 42.90	N/D	80.1 ± 28.9	40.0–150.0
Total Bilirubin (mg/dL)	0.50 ± 0.20	N/D	0.40 ± 0.30	0.20–1.20
Indirect Bilirubin (mg/dL)	0.23 ± 0.09 **	N/D	0.15 ± 0.06	0.01–0.90
Direct Bilirubin (mg/dL)	**0.31 ± 0.13 ****	N/D	0.13 ± 0.10	0.00–0.30
LDH (IU/L)	210.6 ± 67.6	N/D	203.3 ± 54.3	125.0–220.0
CPK total (IU/L)	84.4 ± 58.0	N/D	80.2 ± 63.1	30.0–200.0
Amylase (IU/L)	120.0 ± 39.6	N/D	63.9 ± 18.5	20.0–160.0
Vitamin-D (ng/mL)	**11.6 ± 6.5 ***	N/D	20.5 ± 4.9	30.0–100.0
hs CRP (mg/L)	**12.5 ± 10.2 ****	N/D	1.8 ± 1.0	0.0–5.0

Values are presented as mean ± SD. Bold: pathological values; * *p* < 0.05 vs. controls; ** *p* < 0.01 vs. controls; # *p* < 0.05 pre- vs. post-HD. ALP, alkaline phosphatase; CPK, creatine phosphokinase, γGT, gamma-glutamyl transferase; EpO, erythropoietin; fHb, plasma hemoglobin, Hb, hemoglobin; Hct, hematocrit; hsCRP, high sensitivity C-Reactive Protein; LDH, lactate dehydrogenase; MCH, mean corpuscular hemoglobin; MCHC, mean corpuscular hemoglobin concentration; MCV, mean corpuscular volume; MPV, mean platelet volume; PLT, platelets; PDW, platelet distribution width; PTH, parathormone; RBC, red blood cells; RDW, RBC distribution width; SGOT, serum glutamyl oxalate transaminase; SGPT, serum glutamyl pyruvate transaminase; TIBC, total iron-binding capacity; WBC, white blood cells.

**Table 2 biomolecules-11-01309-t002:** Secondary hemostasis screening tests, coagulation, fibrinolytic and inhibitory proteins.

Characteristics	Patients (*n* = 32)		Controls (*n* = 15)	Normal Range
	Pre-HD	Post-HD		
Secondary hemostasis screening test				
aPTT (s)	**56.7 ± 32.1 ****	35.5 ± 7.0 **^#^	29.7 ± 3.7	<36
D-Dimers (µg/L)	**661.2 ± 438.6 ****	**694.8 ± 425.3 ****	272.7 ± 140.2	<500
Fibrinogen (mg/dL)	**411.8 ± 92.4 ****	**464.5 ± 112.7 **^#^**	328.4 ± 79.6	180–350
PT/INR	**1.4 ± 0.7 ****	1.2 ± 0.4 **^#^	0.98 ± 0.04	<1.2
Coagulation system				
Factor II (%)	79.0 ± 19.3 **	81.4 ± 21.4 **	107.9 ± 13.2	60–120
Factor V (%)	85.7 ± 23.6 **	106.0 ± 19.2 ^#^	104.9 ± 17.2	60–120
Factor VII (%)	105.9 ± 37.0	110.8 ± 39.2	116.0 ± 19.8	60–120
Factor VIII (%)	**148.8 ± 79.9**	**247.1 ± 85.5 **^#^**	123.3 ± 17.6	60–140
Factor IX (%)	115.2 ± 43.4	**123.0 ± 39.9**	115.9 ± 17.9	60–120
Factor X (%)	77.1 ± 27.0	82.0 ± 35.1	89.1 ± 15.8	60–120
Factor XI (%)	100.3 ± 49.6	**121.3 ± 50.6**	112.6 ± 16.1	60–120
Factor XII (%)	87.1 ± 24.2 **	108.1 ± 22.3 ^#^	109.9 ± 26.9	60–120
Factor XIII (%)	**145.9 ± 12.1 ****	**149.8 ± 11.0 ****	101.1 ± 11.9	60–140
RiCof (%)	**160.6 ± 53.6**	**191.8 ± 72.4 **^#^**	122.7 ± 34.1	60–140
vWF (%)	**170.4 ± 58.0 ****	**216.8 ± 74.7 **^#^**	110.0 ± 20.8	60–140
Fibrinolytic system				
PAI (U/mL)	1.9 ± 1.2 **	1.6 ± 1.1 **	0.8 ± 1.1	<4
Plasminogen (ng/L)	100.5 ± 15.3	112.8 ± 17.4 **^#^	108.1 ± 8.8	80–120
Inhibitory system				
Antithrombin III (%)	96.9 ± 13.8 **	105.6 ± 14.5 ^#^	122.3 ± 10.1	80–120
Protein C (%)	80.8 ± 18.8 **	91.6 ± 23.1 **^#^	110.3 ± 14.5	70–140
Protein S activity (%)	89.6 ± 23.3	99.3 ± 33.9	91.9 ± 16.5	60–130
Protein Sf (%)	95.6 ± 69.9 **	92.6 ± 27.4 **^#^	109.1 ± 26.3	60–140
ADAMTS-13				
ADAMTS13 activity (%)	**<5 ****	**<5 ****	57.3 ± 4.8	17–63
ADAMTS13 Inhibitor (%)	**81.4 ± 25.1 ****	**86.6 ± 11.5 **^#^**	12.0 ± 5.0	<30

Values are presented as mean ± SD. PT-INR, Prothrombin time—International Normalized Ratio; aPTT, Activated Partial Thromboplastin Time; RiCof, Platelet Ristocetin Cofactor; Protein Sf, Protein S Antigen free; PAI, Plasminogen Activator Inhibitor; ADAMTS, A Disintegrin, and Metalloproteinase with Thrombospondin motifs. Bold: pathological values, ** *p* < 0.01 vs. controls, ^#^ *p* < 0.05 pre- vs. post-HD.

## Data Availability

All data presented in this study are available upon request.

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
