# Peer review of "Coagulation Abnormalities in Renal Pathology of Chronic Kidney Disease: The Interplay between Blood Cells and Soluble Factors"

_biomolecules, 2021, doi:10.3390/biom11091309_

Round 1
Reviewer 1 Report
Overall, the manuscript is fine but there are few comments to be addressed before publication. The results are interesting. The authors provide evidence for the complex coagulation phenotype in end-stage renal disease (ESRD) patients presenting a variety of biomarkers. Few concerns to be addressed are the following: 1) Which are the exclusion criteria used for controls and which is the reason that diabetic ESRD patients were excluded from this study, as diabetic nephropathy is the most prevalent cause of ESRD. 2) Considering the important role of platelet microvesicles in normal hemostasis, why the authors did not study the possible abnormalities in platelet vesiculation of the patients? 3) In Figure 4b, it would be valuable to add an extra rectangle in order to emphasize to the significant correlations of uremic toxin urea with the soluble coagulation factors and RBCs characteristics.Author Response
Please see the attachment.

Reviewer 2 Report
The authors focused on coagulation abnormality in patients with chronic kidney disease. These patients commonly have blood coagulation disorders. The coagulation process involves the participation of the platelets, vascular endothelium, coagulation system, anticoagulant system and fibrinolytic system. Most coagulation test methods reflect changes in a particular blood coagulation step but have difficulty completely verifying the entire coagulation process in patients with CKD. The important fact is that, hemodialysis seems to augment the prothrombotic potential. This is a very interesting manuscript. However, if it was a smaller group of patients, it would be appropriate to perform the study on a larger number of patients.
Some parts of the manuscript need to be corrected and supplemented in order for this manuscript to be published.
Page 13, 392-395 Discussion: In the study, the authors observed hyperfibrinogenemia, which can lead to thrombotic complications. However, it should be noted that fibrinogen deficiency can lead to thrombosis. Various mechanisms explain the risk of thrombosis in patients with congenital fibrinogen disorders, including elevated levels of thrombin due to the defect in binding fibrinogen, altered strength, structure, and stability of the fibrin clot, as well as impaired fibrinolysis. This is necessary to state and then cite the manuscript in which it was published. Int J Hematol. 2020 Jun;111(6):795-802. doi: 10.1007/s12185-020-02842-9, doi: 10.1016/j.ajem.2010.05.016., https://doi.org/10.5858/2002-126-1387-DAT.
Page 13, 392-395: The manuscript describes hyperfibrinolysis, the authors should state that fibrinolysis is a highly regulated process, starting with fibrin formation and the activation of the tissue plasminogen activator (t-PA) on plasminogen-binding sites. They should cite the manuscript in which it was published: Biomedicines 2020, 8(12), 605; https://doi.org/10.3390/biomedicines8120605, doi: 10.1016/j.blre.2014.09.003
The methodical part is extensive and precisely written. The study included 32 patients and 15 controls. I would like to commend the use of individual laboratory methods.
Figures and tables in the text are very clearly written.
I have to say that in manuscript is overall 56 references there are only 22 references newer than 5 years old.
Reviewer 3 Report
Pavlou, E.G., Georgatzakou, H.T., Fortis, S.P., Tsante, K.A., Tsantes, A.G., Nomikou, E.G., Kapota, A.I., Petras, D.I., Venetikou, M.S., Papageorgiou, E.G., Antonelou, M.H., Kriebardis, A.G. Coagulation abnormalities in renal pathology of chronic kidney disease: the interplay between blood cells and soluble factors. Biomolecules. 2021. Submitted.
Summary: This is a well written article seeking to investigate the changes in end stage renal disease hemostasis due to alterations of blood cells and coagulation factors. The authors compared blood analysis results from healthy patients and those with ESRD both pre- and post-dialysis. They address the apparent dichotomy between the increased bleeding risk present in ESRD patients with the thrombogenic risk associated with hemodialysis. As such they are the first study to address crosstalk between soluble coagulation factors and blood cells. Their results reveal several correlations of clinical interest and identify ADAMTS-13 inhibitor as an important regulator of unknown origin. The manuscript should be accepted but would be better with one major revision.
Major Revision:
- Figure #4 could be significantly improved in the diagram and legend. Specifically, statistical significance is indicated as, black is significant and red is not significant. It is difficult to distinguish which lines are red and which are black. The sentence, “The shortest the edge, the higher the r value between the two interrelated nodes” is unclear. Not sure what the “shortest edge” is. The description of the boxed regions and significance is unclear. Since the correlative analysis in Figure 4 has a central role in the manuscript it may be useful to highlight in a more significant manner which connections the authors feel of greatest importance and improving the general aesthetic of the figure may be helpful.
Minor Points:
- As the unknown origin of ADAMTS-13 inhibitor is a point of note in the abstract, it should be addressed more completely in the authors’ discussion and/or conclusion sections.
- Starting the sentence on line 409 with “Obviously” seems inappropriate. If it is obvious then it hardly needs to be mentioned and presumes an understanding from the audience which may not be present.

Round 2
Reviewer 2 Report
The presented manuscript has been corrected in response to the suggestions. The authors have followed the recommendations of the reviewer. After the revision, the provided data and the addition of the results became more clear. I would like to thank the authors for resubmitting the manuscript and explaining the obscure points from the previous version.